# Performance Characteristics of Next-Generation Sequencing for the Detection of Antimicrobial Resistance Determinants in *Escherichia coli* Genomes and Metagenomes

Ashley M. Rooney,[a,b] Amogelang R. Raphenya,[c,d,e] (ID) Roberto G. Melano,[a,f] Christine Seah,[f] Noelle R. Yee,[b] Derek R. MacFadden,[g] (ID) Andrew G. McArthur,[c,d,e] (ID) Pierre H. H. Schneeberger,[a,b,h] Bryan Coburn[a,b,i]

aDepartment of Laboratory Medicine and Pathobiology, Faculty of Medicine, University of Toronto, Toronto, Ontario, Canada
bUniversity Health Network, Division of Infectious Diseases and Toronto General Hospital Research Institute, Toronto, Ontario, Canada
cDavid Braley Centre for Antibiotic Discovery, McMaster University, Hamilton, Ontario, Canada
dMichael G. DeGroote Institute for Infectious Disease Research, McMaster University, Hamilton, Ontario, Canada
eDepartment of Biochemistry and Biomedical Sciences, McMaster University, Hamilton, Ontario, Canada
fPublic Health Ontario Laboratory, Toronto, Ontario, Canada
gOttawa Hospital Research Institute, Ottawa, Ontario, Canada
hDepartment of Medical Parasitology and Infection Biology, Swiss Tropical and Public Health Institute, University of Basel, Basel, Switzerland
iDepartment of Medicine, Faculty of Medicine, University of Toronto, Toronto, Ontario, Canada

**ABSTRACT** Short-read sequencing can provide detection of multiple genomic determinants of antimicrobial resistance from single bacterial genomes and metagenomic samples. Despite its increasing application in human, animal, and environmental microbiology, including human clinical trials, the performance of short-read Illumina sequencing for antimicrobial resistance gene (ARG) detection, including resistance-conferring single nucleotide polymorphisms (SNPs), has not been systematically characterized. Using paired-end 2 × 150 bp (base pair) Illumina sequencing and an assembly-based method for ARG prediction, we determined sensitivity, positive predictive value (PPV), and sequencing depths required for ARG detection in an *Escherichia coli* isolate of sequence type (ST) 38 spiked into a synthetic microbial community at varying abundances. Approximately 300,000 reads or 15× genome coverage was sufficient to detect ARGs in *E. coli* ST38, with comparable sensitivity and PPV to ~100× genome coverage. Using metagenome assembly of mixed microbial communities, ARG detection at *E. coli* relative abundances of 1% would require assembly of approximately 30 million reads to achieve 15× target coverage. The minimum sequencing depths were validated using public data sets of 948 *E. coli* genomes and 10 metagenomic rectal swab samples. A read-based approach using *k-mer* alignment (KMA) for ARG prediction did not substantially improve minimum sequencing depths for ARG detection compared to assembly of the *E. coli* ST38 genome or the combined metagenomic samples. Analysis of sequencing depths from recent studies assessing ARG content in metagenomic samples demonstrated that sequencing depths had a median estimated detection frequency of 84% (interquartile range: 30%–92%) for a relative abundance of 1%.

**IMPORTANCE** Systematically determining Illumina sequencing performance characteristics for detection of ARGs in metagenomic samples is essential to inform study design and appraisal of human, animal, and environmental metagenomic antimicrobial resistance studies. In this study, we quantified the performance characteristics of ARG detection in *E. coli* genomes and metagenomes and established a benchmark of ~15× coverage for ARG detection for *E. coli* in metagenomes. We demonstrate that for low relative abundances, sequencing depths of ~30 million reads or more may be required for adequate sensitivity for many applications.

**KEYWORDS** antimicrobial resistance gene detection, whole genome, microbiome, resistome, sequencing, metagenome

**Ad Hoc Peer Reviewer** (ID) Xian-Zhi Li, Health Canada

Address correspondence to Bryan Coburn, bryan.coburn@uhn.ca, or Pierre H. H. Schneeberger, pierre.schneeberger@swisstph.ch.

The authors declare no conflict of interest.

Increasing throughput and decreasing costs of DNA sequencing have made whole genome and metagenomic sequencing accessible for antimicrobial resistance detection on a broad scale. This technology is a useful epidemiological tool capable of tracking multiple antimicrobial resistance genes (ARGs) in a single strain (1, 2), and there are increased efforts to correlate isolate genotype with phenotypic resistance (3, 4). The "resistome" (5) is the total genetic content of the microbiome with the potential to confer resistance to antibiotics, and there has been significant interest in characterizing the ARG content in the environment (6–8), humans (9, 10), and other mammals (11, 12). Clinical trials investigating microbiome-based interventions as well as antibiotic efficacy have also included antimicrobial resistance potential in the gut microbiome as an outcome (13–15).

Multiple factors may influence the accuracy of ARG detection from genomes, including the sequencing depth, the quality of the sequencing data, the bioinformatic tools used, as well as the reference database selected for ARG prediction (16). Numerous bioinformatic tools have been developed to tackle ARG detection from whole genomes and metagenomes, which generally include either assembly-based or read-based approaches. Using an assembly-based approach, ARGs can be predicted from genomes and metagenomes by performing *de novo* assembly (assembly without a reference) of the raw sequencing reads into contiguous sequences (contigs) and aligning the contigs to an ARG reference database. In contrast, a read-based approach directly aligns the raw genomic or metagenomic sequences to an ARG reference database for ARG prediction (17). Currently, there is no standard bioinformatic approach (17) as there are relatively few comparisons made between these two methods with a lack of a gold standard where all resistance determinants are known, which does not allow for a fair evaluation of either approach (18, 19).

It is important to estimate the sequencing depth needed per sample as low sequencing depths may not capture the entire genomic content where the amount of sequencing reads required is dependent on the goal of the study (20). One study found that a minimum of 500,000 sequencing reads provided similar microbiome diversity and functional profile to deep sequencing (21), while another study found that a minimum of 15–20× genome coverage was sufficient to detect most ARGs in 111 *Salmonella enterica* isolates (18). To our knowledge, there are no recommendations for optimal sequencing depths required to detect ARGs and resistance-conferring single nucleotide polymorphisms (SNPs) in metagenomic samples, while the performance characteristics of different sequencing depths using common bioinformatic tools for genomes and metagenomes have not been well established.

In this study, we used the Resistance Gene Identifier (RGI) and the Comprehensive Antibiotic Resistance Gene Database (CARD) (22) with an assembly-based approach to assess the limits of detection, sensitivity, and positive predictive value (PPV) of sequencing to detect known antimicrobial resistance determinants including ARGs and alleles, as well as resistance-conferring SNPs in a multidrug-resistant *Escherichia coli* isolate that represented varying abundances in a complex metagenome. The sequencing depths for the detection of all ARGs were validated in 948 *E. coli* genomes. For metagenomic samples, sequencing depths were validated for the detection of *vanA* in a publicly available data set of 10 rectal swab samples with a range of *vanA*-carrying *Enterococcus* relative abundances. We highlight the importance of maintaining minimum target genome coverage to detect ARGs when the target organism is at varying relative abundances in a metagenomic sample, and provide an estimate of minimum required sequencing depths of target organisms to maintain adequate sensitivity.

## RESULTS

**Antimicrobial resistance determinant detection in *Escherichia coli* ST38.** To ensure sequencing depth was not a limiting factor in ARG detection, approximately 136 million 2 × 150 bp (base pair) reads (~6,800× genome coverage) were generated for the *E. coli* ST38 genome. As our goal was to determine minimum sequencing depths needed to detect ARGs, subsamples were examined at 5,000,000 (~250×), 1,000,000 (~50×),

500,000 ($\sim$25$\times$), 300,000 ($\sim$15$\times$), 250,000 ($\sim$12.5$\times$), 200,000 ($\sim$10$\times$), 150,000 ($\sim$7.5$\times$), 100,000 ($\sim$5$\times$), 50,000 ($\sim$2.5$\times$), and 10,000 ($\sim$0.5$\times$) read pairs to simulate sequencing at lower depths. Reads were then assembled into contigs and ARGs predicted using RGI *main* (see Materials and Methods), with the ARGs and associated detection frequencies across subsamples in Data Set S1. ARGs detected with $\geq$90% detection frequency were considered high confidence genes, whereas those detected with $\leq$50% detection frequency were considered low confidence genes.

A sequencing depth of 300,000 reads or approximately 15$\times$ genome coverage was sufficient to detect the known $bla_{CTX-M-15}$, and *parC* and *gyrA* variants as well as 69 other ARGs with a mean detection frequency of 99.9% $\pm$ 0.5% (standard deviation), where additional sequencing depth did not increase the total number of ARGs detected even by 5,000,000 reads (Fig. 1a). Other ARGs included 3 different beta-lactamases ($bla_{TEM-1}$, $bla_{OXA-1}$, and $bla_{AmpC}$), 5 unique aminoglycoside transferases, and 46 distinct efflux-associated genes of which 41.3% (19/46) are regulatory (Fig. 1b). There were ARGs detected at $\leq$50% detection frequency across all sequencing depths except at 500,000 reads, where no ARGs were detected at these low detection frequencies (Fig. 1a).

To demonstrate how sequencing depth affects the performance of ARG detection, sensitivity and PPV were calculated across subsamples. Specificity was not used as a metric to assess performance, due to the high number of true negatives, which would inflate specificity. A depth of 300,000 reads ($\sim$15$\times$ coverage) performed similarly to 1 million reads for sensitivity (1.00 $\pm$ 0.00 versus 1.00 $\pm$ 0.00, Fig. 1c) and PPV (mean = 1.00 $\pm$ 0.00 versus 1.00 $\pm$ 0.00, Fig. 1d) with low false negatives (0.09 $\pm$ 0.29, Fig. 1c) and false positives (0.02 $\pm$ 0.14, Fig. 1d) (mean and standard deviation).

As BLAST is a highly sensitive alignment tool (23), we hypothesized that BLAST may improve ARG detection frequencies at lower sequencing depths compared to DIAMOND. The ARGs and associated detection frequencies using BLAST are in Data Set S1. Overall, more ARGs were predicted using BLAST across all subsamples (Fig. S1c in the supplemental material). But most of the additional ARGs predicted were low confidence (Fig. S1e) and unique to BLAST (Fig. S1f). A total of 72 ARGs achieved high confidence by 300,000 reads (Fig. S1a), which is consistent with results using DIAMOND (Fig. S1b). Between BLAST and DIAMOND, the high confidence genes predicted at subsamples $\geq$300,000 reads were similar in number (approximately 72 genes were predicted by both methods) as well as annotation (Fig. S1d and S1f).

**Resistance-conferring SNP detection in *E. coli* ST38.** A total of 6 ARG variants were detected including the *E. coli gyrA* and *parC* variants conferring resistance to fluoroquinolones as well as *E. coli* EF-Tu mutants conferring resistance to pulvomycin, *E. coli cyaA* with a mutation conferring resistance to fosfomycin, *E. coli glpT* with a mutation conferring resistance to fosfomycin, and a PBP3 conferring resistance to beta-lactam antibiotics. Of the 6 ARG variants and associated SNPs evaluated, 200,000 assembled reads were sufficient to detect 6/6 with high confidence (Fig. 1e). Variants predicted with RGI's protein variant model had similar detection frequencies across subsamples as their associated SNPs, except for *gyrA* with the resistance-conferring SNPs D87N and S83L, where these SNPs were detected 1–2% less frequently than the variant itself at subsamples 50k to 150k. Instead S83L or D87N was detected individually (Data Set S2).

**Validation of 15X coverage across *E. coli* isolate genomes.** The 300,000 read depth (15$\times$ coverage) threshold for ARG detection was validated across 948 *E. coli* isolates (24). The isolates were previously sequenced to an average of 100$\times$ coverage using 150-bp paired-end Illumina sequencing. A 300,000 read subsample from each isolate was performed, and the ARGs predicted at 300,000 assembled reads using RGI *main* were compared to the ARGs predicted at the original sequencing depth to calculate sensitivity, PPV, and F1 score for each isolate.

Across the *E. coli* isolate set, a total of 322 unique ARGs were observed. Table S1 outlines the detection and performance characteristics of each gene. The overall performance of the 300,000 read depth is summarized in Fig. 1f. The F1 score was 1 for 658/948 (69.4%) isolates. There were 290/948 (30.6%) isolates with a F1 score of <1, where 228/290 (78.6%) isolates had an F1 score between 0.99–0.98, 49/290 (16.9%)

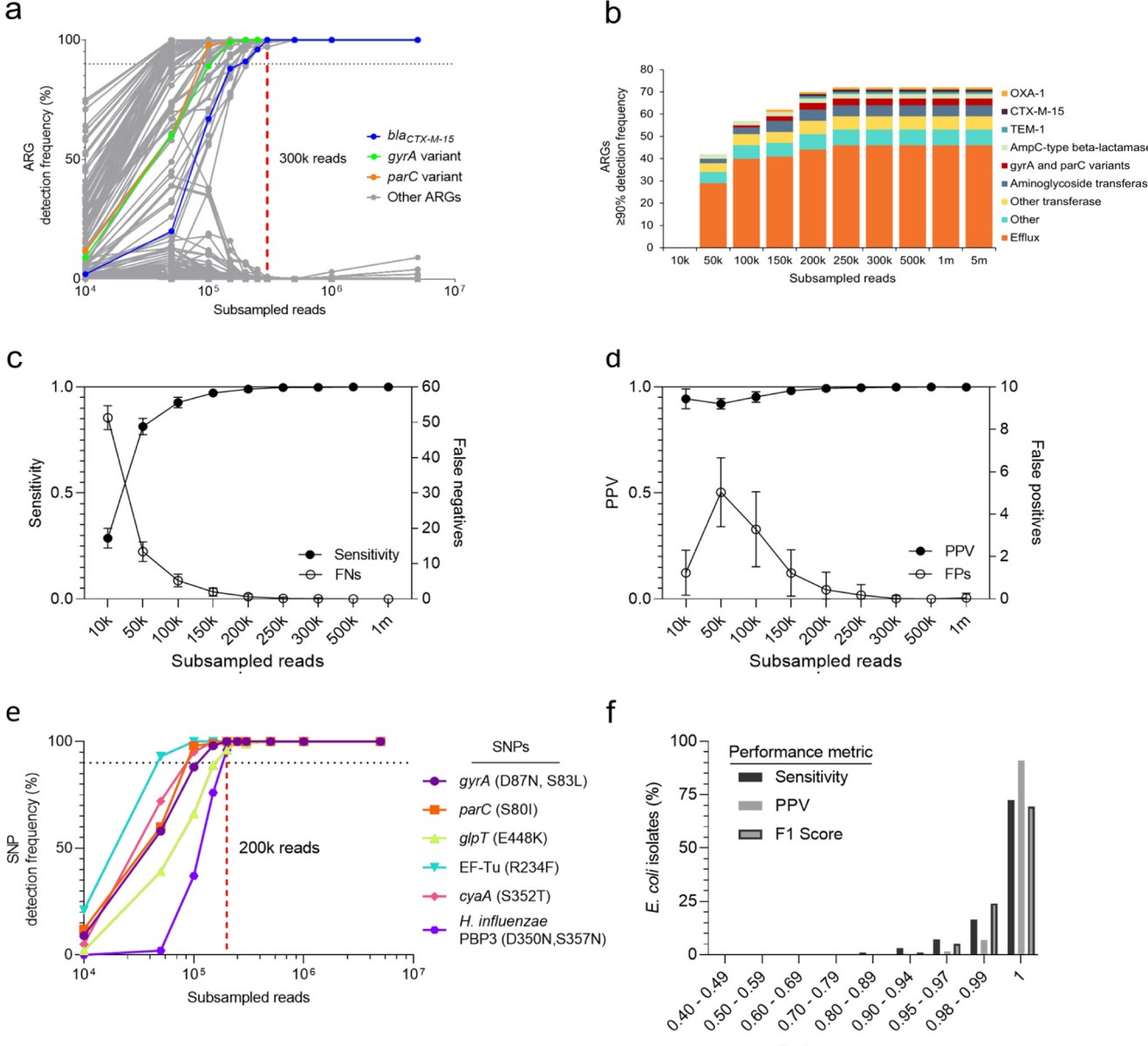

**FIG 1** Genomic antimicrobial resistance determinant detection. (a) ARG detection frequencies across subsamples in *Escherichia coli* ST38. Individual dots represent a single ARG and are connected by lines to demonstrate trends in detection across subsamples. $bla_{CTX-M-15}$, *gyrA*, and *parC* variants are highlighted as previously identified resistance determinants for this strain. (b) Histogram of the number of unique ARGs with ≥90% detection frequency summarized by categories detected across subsamples in *E. coli* ST38. (c–d) Performance of ARG classification across subsamples in *E. coli* ST38 including (c) sensitivity and false negatives, and (d) positive predictive value (PPV) and false positives (FPs). The mean and standard deviation are plotted. (e) Protein variants and associated SNP(s) detection frequencies across subsamples in *E. coli* ST38. (f) A distribution of the percentage of *E. coli* isolates (n = 948) by ARG detection performance of 300,000 reads compared to 100× genome coverage. Performance is measured by sensitivity, PPV, and F1 score. In a and e, the horizontal dotted line marks 90% detection frequency. The red vertical dashed line marks the subsample at 300,000 reads (a) and 200,000 reads (e).

had an F1 score between 0.95–0.97, 11/290 (3.8%) had an F1 score between 0.90–0.94, and the remaining isolates had F1 scores of 0.89 and 0.65. Of the 290 isolates with F1 score <1, 84 (29.0%) had a PPV of <1 and 261 (90.0%) had a sensitivity of <1. For the isolates with a PPV of <1 (*n* = 84), 65 isolates had 1 false positive, 12 isolates had 2 false positives, 3 isolates had 3 false positives, and the remaining 4 isolates had 5, 7, 16, and 70 false positives, respectively. For the isolates with a sensitivity of <1, the median number of false negatives was 1 (range 1–15).

Of the 322 unique ARGs, 21 genes (6.5%) were classified as true positive for all 948 isolates, of which 90.5% (19/21) were efflux-associated genes. The top three ARGs that

contributed the most false negatives were *APH(6)-ld* (*n* = 25), *sul2* (*n* = 23), and *mphA* (*n* = 23), with a sensitivity of approximately 93% for all three genes respectively, while the top three genes that contributed the most false positives were *bla*$_{OXA-320}$ (*n* = 13), *aadA* (*n* = 6), and *bla*$_{OXA-140}$ (*n* = 6) (Table S1).

**Detection of *E. coli* ST38 ARGs and SNPs at a range of relative abundances in a metagenomic sample.** To assess the effect *of E. coli* ST38 relative abundance on ARG detection in a multispecies metagenome, the DNA of *E. coli* ST38 and a 34-species cultivated microbial consortium, where each isolate was subject to antibiotic susceptibility testing and included in the consortium if phenotypically susceptible to a range of antibiotics (25), were combined to create metagenomic samples where *E. coli* ST38 represented approximately 90%, 50%, 10%, and 1% of the total metagenome. Of note, the microbial consortium also included an antibiotic-susceptible *E. coli* isolate, accounting for approximately 2% of the total microbial abundance. Based on the sequencing limit of detection of 300,000 reads in the single *E. coli* ST38 isolate (100% relative abundance), we estimated that at 90%, 50%, 10%, and 1% relative abundance, 333,333, 600,000, 3,000,000, and 30,000,000 2 × 150 bp reads, respectively, would be required to detect reference-based ARGs (*n* = 20) and protein variant-associated SNPs (*n* = 6) contributed by the *E. coli* ST38 isolate with ≥90% detection frequency. Reads were assembled into metagenomic contigs and ARGs predicted using RGI *main* with its *low_quality* flag (see Materials and Methods). The reference-based ARGs and SNPs detected in each of the four metagenomic samples across subsamples are summarized in Fig. 2c.

As the *E. coli* ST38 relative abundance decreased, the number of reads necessary to detect the reference-based ARGs (Fig. 2a and c) and SNPs (Fig. 2b and c) increased. The detection rate approximated our expectations at relative abundances of >1% (Fig. 2a to c). For the combined sample containing *E. coli* ST38 at 1% relative abundance, 90% (18/20) of the reference-based ARGs and 67% (4/6) of the SNPs had a detection frequency of ≥90% with ≤30,000,000 reads (Fig. 2c). The ARG annotated as *ugd* had a detection frequency 50% (Fig. 2a and c), the *cyaA* SNP (S352T) had a detection frequency of 40%, and the *gyrA* SNPs (D87N, S83L) were not detected at 30,000,000 reads (Fig. 2b and c).

Across the four metagenomic samples, the ARG and SNPs annotated as CTX-M-15 and the protein variant PBP3 (SNPs: D350N, S357N), respectively, required the most sequencing information to detect them at each of the detection frequency cutoffs compared to the other ARGs and SNPs.

**Validation of 15× coverage in metagenomic samples.** From a public data set of 10 rectal surveillance swabs that were vancomycin-resistant *Enterococcus* positive by culture and *vanA* positive in 9/10 swabs by Illumina sequencing (26), we validated 15× *Enterococcus* genome coverage for the detection of *vanA*. The study authors performed 2 × 75 bp sequencing and achieved a mean 9.1 million reads (range: 5.7–15 million reads), post quality filtering and removal of human reads. The rectal swab samples had a range of *Enterococcus* relative abundances (median: 0.10; range: 80%–0.02%) and genome coverages (median: 21×; range: 375×–0.07×) (Fig. 2d). Rectal swab number 8 had the highest *Enterococcus* relative abundance of 80%, and due to the large number of sequencing reads (12.5 million), had the largest estimated target genome coverage of ∼375×. Rectal swab number 4 had the lowest *Enterococcus* relative abundance (0.02%) and 9.1 million sequences, which resulted in a target genome coverage of ∼0.07× for this sample (Fig. 2d).

To assess whether a minimum of 15× target genome coverage is sufficient to detect *vanA* in the rectal swab metagenomes, reads were sampled 10 times at each subsampling depth to achieve a range of target genome coverages from 0.5× to 15× to determine *vanA* detection frequency. As above, reads were assembled into metagenomic contigs and ARGs predicted using RGI *main* with its *-low_quality* flag. The results of this analysis are displayed in Fig. 2e. To achieve 100% detection frequency of the *vanA* gene across rectal swab samples, 5 rectal swab samples required *Enterococcus* genome coverage of less than 5× (rectal swabs 1, 5–7, and 10), while 2 required at least 15× coverage (rectal swabs 3 and 8). At 15× *Enterococcus* genome coverage, *vanA* was detected in 10/10 bootstraps for all samples that

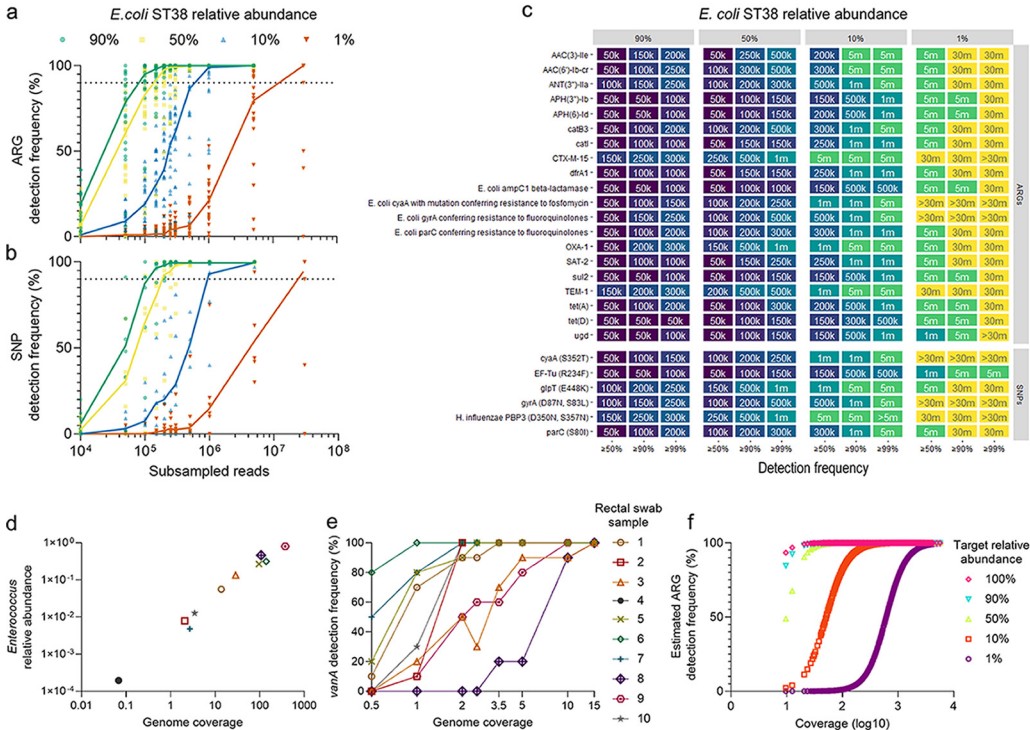

**FIG 2** Metagenomic antimicrobial resistance determinant detection. (a–c) Reference-based ARG (*n* = 20) and SNP (*n* = 6) detection frequencies in four metagenomic samples with *Escherichia coli* ST38 relative abundances of 90%, 50%, 10%, and 1%. (a–b) Individual dots represent a single ARG or SNP. Trend lines for each of the four metagenomic samples are plotted through the median detection frequency at each subsample. The horizontal dotted line marks 90% detection frequency. (c) For reference-based ARGs or SNPs in each metagenomic sample, the minimum subsample that falls within the detection frequency cutoff (*x* axis) is plotted. (d–e) Detection of *vanA* in rectal swab samples positive for vancomycin-resistant *Enterococcus* from a public data set. (d) *Enterococcus* relative abundance by total genome coverage; each rectal swab sample is represented by an icon. (e) *vanA* detection frequency across genome coverages for each rectal swab sample. Rectal swab sample 4 is not plotted, as *vanA* was not detected with the total number of sequences available. (f) Estimated ARG detection frequency by coverage of a hypothetical target organism at a range of relative abundances.

had adequate sequencing depth for subsampling. Rectal swab number 4 did not have enough reads to achieve 0.5× *Enterococcus* genome coverage, and *vanA* was not detected when we analyzed all reads available, which is consistent with the authors' published findings that describe their inability to detect *vanA* using paired-end Illumina sequencing (26).

**Estimates of the sensitivity of sequencing depth for ARG detection in published data sets.** Recent publications assessing ARG content in metagenomic samples may not have achieved optimal sensitivity for ARG detection if they were to use a contig assembly approach. As shown, the relative abundance of the target organism affects the sensitivity to detect ARGs in a metagenomic sample. Sequencing depths and read length data were gathered from three recently published studies that reported ARGs in metagenomic samples to determine coverage of a hypothetical organism at a range of relative abundances. Study 1 compared the resistome of 1,174 gut and oral samples from previously published sources distributed by country (9); 1,132/1,174 samples for which complete read length data (excluding 42/1,174, 3.6%) were available were included in the analyses. Study 2 performed a longitudinal assessment of the gut microbiota and resistome of healthy veterinary students exposed to a Chinese swine farm environment. A total 63 metagenomic samples were sequenced that consisted of human stool and environmental samples (6). Study 3 was conducted in Denmark and evaluated the changes in the gut microbiota composition and resistome of 12 healthy male volunteers before and after antimicrobial exposure (10). A total of 57 stool samples were subject to metagenomic analyses. Studies 1 and 3 used a read-based approach for ARG prediction, while Study 2 used an assembly-based approach.

The ARG detection frequency was estimated from the published sample sequencing depths for a hypothetical target organism at a range of potential relative abundances.

Assuming detection frequency is related to sequencing sensitivity, genome coverage was calculated as an estimate of sequencing depth and interpolated detection frequency values from a sigmoidal curve fit to the *E. coli* ST38 $bla_{CTX-M-15}$ detection frequency data. As the relative abundance of the hypothetical target organism decreased, more sequencing effort was required to achieve 100% estimated detection frequency of all ARGs (Fig. 2f). Most published samples had achieved ≥95% estimated detection frequency for all ARGs for a target organism at relative abundance of 100% (1,251/1,252; 99.9%), 90% (1,250/1,252; 99.8%), and 50% (1,247/1,252; 99.6%). However, the proportion of samples with at least 90% estimated detection frequency was lower for a target organism relative abundance of 10% (1,090/1,252; 87.1%) and 1% (454/1,252; 36.3%). Additionally, 29.5% (369/1,252) of samples were not sequenced sufficiently to achieve >50% estimated detection frequency for a target organism relative abundance of 1%, where 9.2% (115/1,252) had less than 1% estimated detection frequency (Fig. 2f).

**Read-based antimicrobial resistance determinant detection in *E. coli* ST38 and metagenomes.** As assembly may introduce its own degree of uncertainty into ARG prediction, we next compared a read-based approach (*k-mer* alignment; KMA) to the assembly-based approach to determine if it would improve ARG prediction at lower sequencing depths. Reads (2 × 150 bp) were aligned to CARD reference sequences using KMA within RGI's *bwt* branch (see Materials and Methods). All KMA-predicted ARGs across subsamples in *E. coli* ST38 are outlined in Data Set S3. KMA was more sensitive than contig assembly for detection of reference-based ARGs in the *E. coli* ST38 genome (Fig. 3a) and metagenomes at all *E. coli* relative abundances (Fig. 3b to e). However, KMA had a relatively small impact on read-depth requirements. The minimum sequencing depths needed to detect 18/20 reference-based ARGs with ≥90% detection frequency for *E. coli* ST38 abundances for read-based versus assembly-based approaches at 90%, 50%, 10%, and 1% were 200,000 versus 250,000 reads, 300,000 versus 500,000 reads, 5,000,000 versus 5,000,000 reads, and 30,000,000 versus >30,000,000 reads, respectively (Fig. 3f and Fig. 2c). For resistance-conferring SNPs, KMA detected only 3/6 reference-based SNPs with >90% detection frequency (Fig. 3f and g). The PBP3 with SNPs D350N and S357N were not detected in any metagenomic sample, while the SNPs located in the *gyrA* (D87N, S83L) and *parC* (S80I) genes did not achieve a detection frequency ≥50% even at maximum subsample depths in most metagenomic samples (Fig. 3f).

Lastly, for all metagenomic samples, KMA predicted a greater number of nonreference ARGs (probable false positives) (36.1 ± 6.1; mean and standard deviation) with as little as 50,000 reads compared to assembly at the same depth (2.6 ± 2.1). With increasing read depth, the number of nonreference ARGs predicted by KMA increased to 87.0 ± 5.1 at 5,000,000 reads. The increase in nonreference ARGs with increasing read depth was not observed with an assembly-based method at 5,000,000 reads (3.2 ± 3.1) (Fig. 3h). Thus, there was a significant trade-off between ARG recall and precision using KMA for ARG prediction.

We hypothesized that filtering potentially erroneous ARGs would improve the precision of KMA. Four ARG filtering strategies at a range of cutoff values were evaluated for their effect on the performance of KMA (Fig. S2). The ARG filtering strategies included percent coverage, average depth of coverage, number of completely mapped reads, and the average mapping quality (MAPQ) score. The unfiltered precision and recall were 66% and 93%, respectively. No filtering strategy substantially improved the precision of KMA without affecting recall (Fig. S2). Out of the four strategies, percent coverage achieved the greatest increase in precision at the highest stringency cutoff 100% allele coverage (unfiltered precision: 66%; filtered precision: 76%). However, at this cutoff, the recall decreased from 93% to 79%.

## DISCUSSION

Our goal was to characterize the sequencing performance (sensitivity, PPV, and limits of detection) for the detection of known determinants of resistance in metagenomic samples to inform the use of these approaches in human, animal, and environmental studies. It is axiomatic that sequencing depth affects ARG assay sensitivity in single isolates (16, 18) and within a microbiome (27); however, the performance characteristics

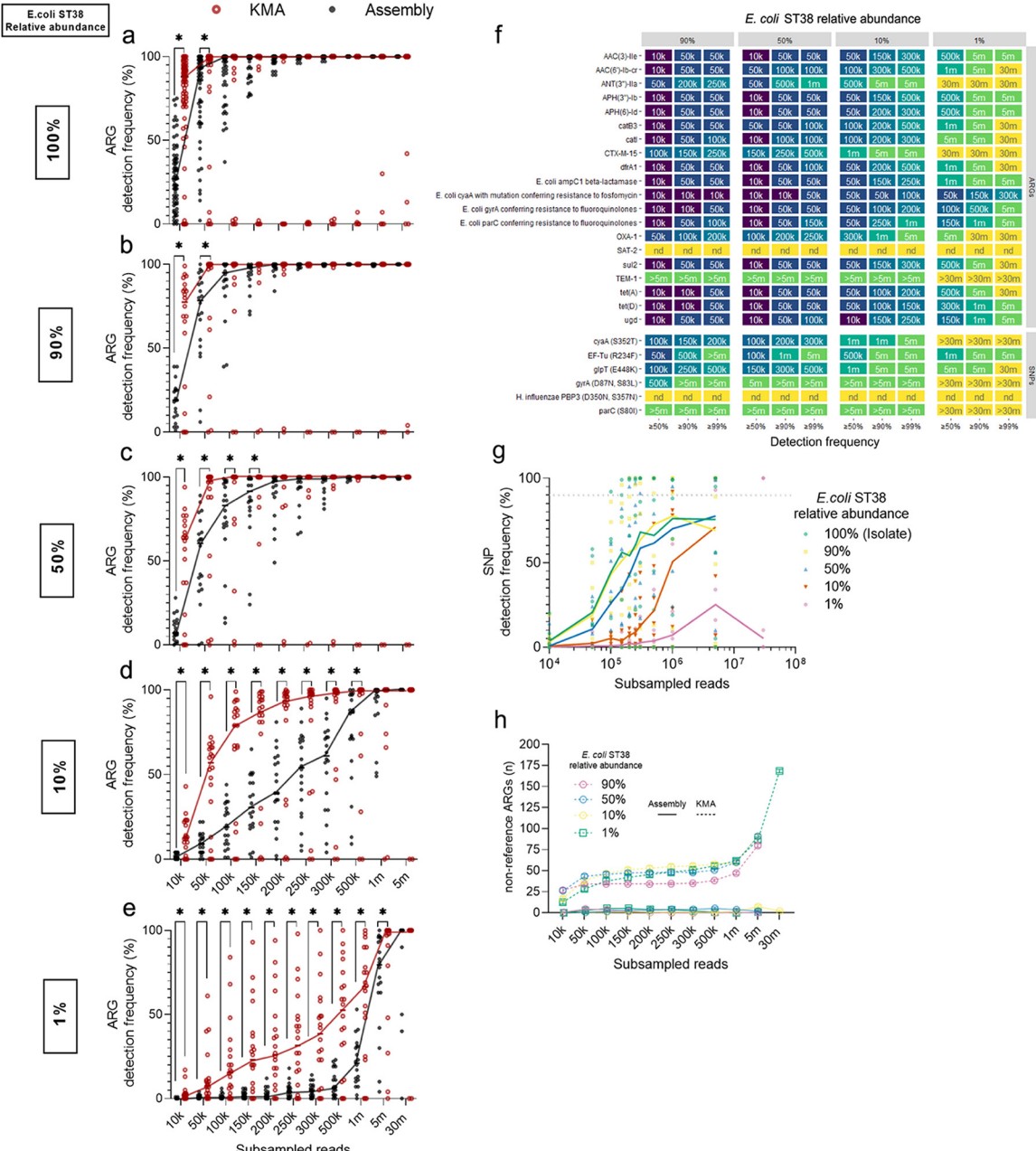

**FIG 3** Performance of a read-based method (KMA) for antimicrobial resistant determinant detection in *Escherichia coli* ST38 and metagenomic samples, compared to assembly. (a–e) A comparison of ARG detection frequencies between subsamples using KMA or assembly. Trend lines are plotted through the median detection frequency at each subsample. Individual points represent single ARGs. Wilcoxon matched-pairs signed-rank test performed at each subsample, *, $P < 0.05$. (a) Reference-based ARG ($n = 72$) detection frequencies in *Escherichia coli* ST38. (b–f) Reference-based ARG ($n = 20$) detection frequencies in four metagenomic samples with *E. coli* ST38 relative abundances of 90% (b), 50% (c), 10% (d), and 1% (e). (f) For reference-based ARGs or SNPs in each metagenomic sample, the minimum subsample that fell within the detection frequency cutoff is plotted. ARGs or SNPs with 0% detection frequency across all subsamples within a sample are indicated as not detected (nd). (g) Reference-based SNP detection frequencies ($n = 6$) using KMA in *E. coli* ST38 (relative abundance 100%), and four metagenomic samples (f–g) with *E. coli* ST38 relative abundances of 90%, 50%, 10%, and 1%. (g) Trend lines are plotted through median detection frequencies at each subsample. Individual points represent individual protein variants and associated SNPs ($n = 6$). The horizontal dotted line marks 90% detection frequency. (h) Nonreference ARGs detected across subsamples in four metagenomic samples with *E. coli* ST38 relative abundances of 90%, 50%, 10%, and 1%. Solid lines or dashed lines are nonreference ARGs detected using contig assembly or KMA, respectively. Mean and standard deviation are plotted.

of sequencing have not been systematically assessed. In published reports, a range of whole genome sequencing depths for single isolates, from $30\times$ coverage up to $100\times$ coverage, are often used to define quality control limits, but these are not considered standard (3, 16, 28, 29). Estimating the coverage of the metagenome required to

ensure high sensitivity is not a new concept (30), but this study precisely quantifies sequencing depths required to detect reference-associated ARGs across a range of relative abundances in mixed metagenomes using standard methods.

Using a *de novo* contig-assembly approach, approximately 15× coverage (300,000, 2 × 150 bp paired-end reads of an estimated 6 Mbp genome) provides similar sensitivity to higher sequencing depths for the detection of ARGs in *E. coli* isolates and is sufficient for detecting SNPs and other resistance genes. Although sequencing depths as low as 0.5 million reads have been proposed to capture the total compositional information of metagenomes (21), greater sequencing depth is required for the detection of ARGs in organisms with low relative abundance, which can require as many as 30 million reads to achieve adequate sensitivity for organisms at a relative abundance of 1%.

For some study purposes, detection of ARGs in low abundance organisms may be critical for study interpretation. Human observational studies have demonstrated that pathogens at both high and low relative abundances in complex gut microbial communities are associated with subsequent infections or death. Dominance of a microbial community by a pathogen is associated with subsequent infection (31–33), but even at relative abundances as low as 1% −0.1%, pathogens detected in stool have been implicated in subsequent bacteremia in hematopoietic stem cell transplant recipients (34), as well as bacteriuria and urinary tract infection (35), indicating that detection of ARGs may be clinically significant even at very low relative abundance thresholds. Based on our estimations, approximately 64% of the samples in recent studies evaluating ARG content in the metagenome are not sequenced at a sufficient depth to detect ARGs in a target organism at 1% relative abundance. Thus, potentially clinically meaningful resistance determinants may not be detected with common sequencing depths such as those analyzed in published studies.

Assembly is time-consuming, requires large amounts of computing power for metagenomic samples, and may also contribute to loss of data (19) as repetitive DNA regions complicate genome assembly resulting in fragmented, short contigs (36) where ARGs may be split between multiple contigs (37). Alternative approaches to assembly such as read alignment (38–41) and kmer-based approaches (19) may require less sequencing information for ARG detection, which is useful for detecting ARGs in low abundance organisms in complex communities. Compared to assembly, KMA did not substantially improve the limit of detection in *E. coli* ST38 or metagenomes, even for a low abundance target, where some genes (e.g., *gyrA* and *parC* SNPs and *bla*$_{TEM-1}$) that had a high detection frequency with assembly, had a low detection frequency with KMA. Although KMA was designed to overcome the problem of mapping sequences to highly redundant gene databases (41), our results suggest that KMA still suffered from the antimicrobial resistance allele network problem where one sequence can map to multiple alleles, which can increase false positives (42). We found that the increased number of false positives detected with KMA could not be filtered without affecting the overall sensitivity of the assay. The differentiation of ARG alleles is important as different alleles, for example mcr-1 versus mcr-9, are associated with different resistance phenotypes (43). Other sequencing technologies such as long-read sequencing provides a promising alternative to short-read sequencing that can overcome the issue of fragmented contig assembly and the potential loss of data (36).

Our approach has the following limitations. A single approach utilizing a widely used sequencing strategy, two bioinformatic pipelines, and one ARG detection platform (CARD) for a single organism (*E. coli*) was modeled. These selections were made to reflect dominant modes of metagenome analysis in a clinically relevant organism to define the "order of magnitude" of depth required for ARG detection from metagenomes, which may not be generalizable to all organisms, community types, or modes of resistance. The microbial consortia of 34 bacterial species used in this study to represent a metagenomic sample, is not comparable to highly diverse sample types such as soil (44). Even in the presence of a susceptible *E. coli* likely carrying wild-type alleles in the microbial consortia, we found that we could detect the *E. coli* ST38 *gyrA* and *parC* protein variants including their associated SNPs provided the minimum sequencing

depth was achieved. However, there were challenges detecting the *gyrA* variant in the combined metagenomic sample where the spiked-in *E. coli* ST38 accounted for 1% of the total sample and the susceptible *E. coli* accounted for 2% of the total sample. A total of 30 million reads allowed for the detection of the mutant *parC* with its associated SNPs, but we did not detect the mutant *gyrA* variant even at this high sequencing depth. Our sample set was not created to directly assess the performance of detecting protein variants (e.g., *gyrA* or *parC*) in the presence of more abundant wild-type alleles.

The metagenomic data set consisting of 10 rectal swab samples representing our metagenomic validation set only focused on the detection of a single ARG, *vanA*. It is possible that other ARGs in similar sample types may require more sequencing depth to detect them.

We chose to spike the *E. coli* ST38 isolate into microbial consortia instead of other representative sample types, because the microbial consortia were prepared rigorously where each isolate included in the mixture underwent antibiotic susceptibility testing and was included if phenotypically susceptible to a range of antibiotics (25). The metagenomic data set of 10 rectal swab samples was chosen because, to our knowledge, it is the only publicly available data set that has combined culture-confirmed vancomycin resistance in a target organism (*Enterococcus* spp.) from metagenomic samples, with paired-end Illumina sequencing data. The culture confirmation of vancomycin resistance and target organism is critical as it provided ground truth to base the sequencing depth recommendations on.

A main limitation of ARG prediction from sequencing data is the chosen database, which can potentially increase false negatives. However, CARD is widely used, updated on an approximately monthly basis, and is representative of known ARG diversity, especially for well-characterized pathogens such as *E. coli* (22). Human metagenomic samples often have human DNA that can account for a large proportion of the total sample, which impacts sequencing strategies (45, 46). An understanding of the total genetic material contributed by human reads prior to sequencing would further inform sequencing effort required to maintain a minimum sequencing depth for AMR gene detection. In addition, sequencing-based approaches may be augmented with bait capture approaches that enrich for ARG target that may increase efficiency and sensitivity (47). We did not directly attempt to connect the ARGs detected in the metagenomic samples to the *E. coli* ST38 using bioinformatic techniques such as contig binning, as it has been shown to produce false negatives for plasmid-borne ARGs (48). DNA manipulation methods such as Hi-C may be useful for linking ARGs to the bacterial hosts (49), but this method would require further validation.

**Conclusions.** As metagenomic sequencing methods are increasingly targets for translational applications for human, animal, and environmental microbiomes as well as antimicrobial resistance research, it is imperative that the performance characteristics of commonly used tools be systematically and carefully characterized. We have quantified sequencing depths needed to detect ARGs in *E. coli* whole genomes and in an *E. coli* isolate ranging from high to low relative abundances in a complex community. A minimum of 15× coverage is needed for the detection of AMR genes in *E. coli* using an assembly-based approach. For metagenomic samples, 15× coverage is also sufficient to detect known ARGs, with the number of 2 × 150 bp sequences ranging from ~333,333 to >30 million to ensure adequate coverage at relative abundances of 90%–1%, which have been implicated in human infection. Currently, sequencing depths reported for metagenomic samples intended for ARG assessment may not be sequenced sufficiently deeply to detect ARGs from bacteria with relative abundances of ~1%. This has implications across a broad range of applications, including clinical trials assessing the impacts of antibiotics on the resistome and antimicrobial resistance surveillance in the environment and animals, where studies may be at risk of under-reporting the burden of antimicrobial resistance. We believe that our analysis provides a robust benchmarking of sequencing effort for metagenomic studies in which detection of resistance is a specified outcome that will allow researchers to calibrate the adequacy of their sequencing efforts for ARG detection in metagenomes.

## MATERIALS AND METHODS

**Sample preparation and sequencing.** From a collection of previously characterized *E. coli* isolates (24), a multidrug-resistant *E. coli* of sequence type 38 with an extended spectrum beta-lactamase (*bla*$_{CTX-M-15}$) and fluoroquinolone resistance-conferring SNPs in *parC* (S80I) and *gyrA* (S83L, D87N) was selected. Briefly, *E. coli* ST38 was cultured from a glycerol stock on LB agar, and a single colony was inoculated into 25 mL of LB broth, which was placed on a shaker incubator (130 rpm) at 37°C for 4 h until media was turbid. Turbid media (25 mL) was transferred to a 50 mL conical tube, subject to centrifugation at 2,500 *g*, the supernatant removed, and the pellet resuspended in 500 μL of LB broth. A description of the microbial consortia preparation was described previously (25). Aliquots of the microbial consortia were stored at −80°C prior to use.

DNA was extracted from thawed microbial consortia (250 μL), and the *E. coli* ST38 isolate in LB broth (250 μL) using the DNeasy PowerSoil kit (Qiagen) and DNA concentration was measured using a Qubit Fluorometer (Thermo Fisher), following the manufacturer's instructions, respectively. *E. coli* and microbial consortia DNA were combined to a final concentration of 20.1 ng/μL, while varying the concentration of *E. coli* so that it approximately represented 90%, 50%, 10%, 1%, 0.1%, 0.01%, 0.001%, and 0.0001% relative to the microbial consortia. Sequencing libraries were prepared using the Nextera DNA Flex kit (Illumina) following the manufacturer's instructions and stored at −20°C. All 10 samples (the *E. coli* ST38 isolate, microbial consortia, and 8 combined samples) were subject to paired-end sequencing at 2 × 150 bp on the NovaSeq 6000 at the Princess Margaret Genomics Centre. Since the minimum sequencing depths needed to detect ARGs in the combined samples where *E. coli* ST38 represented 0.1%, 0.01%, 0.0001%, and 0.0001% were not achieved, these samples were not analyzed.

**Bioinformatic analyses.** All bioinformatic analyses were performed with default settings except where stated. Briefly, from each pair of fastq files, Seqtk (50) v.1.3 was used to subsample *n* number of reads, specifying the *–sample* flag with 100 bootstraps performed (sampling with replacement) for all samples and subsamples, except the combined sample with an *E. coli* ST38 relative abundance of 1% where at the 30 million read subsample 10 bootstraps were performed. In seqtk, the seed parameter, *-s*, *had* a unique number to ensure every bootstrap was a random sampling of reads. Paired-end fastq files (read 1 and read 2) were assessed for quality using FastQC (51) v.0.11.9. The sequence quality and sequence lengths (% less than 20 bp or 35 bp) across subsamples are summarized in Table S2. Neither read trimming nor sequence length cutoffs were applied in the quality filtering process. However, Nextera adapters were removed with Trimmomatic (52) v.0.39. Reads for *E. coli* genomes as well as the microbial consortia and the combined samples were assembled into contigs using SPAdes (53) v.3.13.1, specifying the *–careful* flag, and metaSPAdes (54) v.3.13.1, respectively, using the recommended kmer lengths 21, 33, 55, and 77. Quast (55) v.5.0.2 was performed without a reference, on *de novo* assembled contigs to assess contiguity-based metrics. Fig. S3 summarizes the N50, total number of contigs, and total length of the assembly (bp) across subsamples. Metaphlan2 (56) v.2.9.21 was used to confirm sample taxonomy, including the identity of all *E. coli* isolates and the relative abundance of *Enterococcus* species in the validation sets, respectively.

To predict ARGs from contigs, RGI *main* v.5.1.0 of the CARD v.3.1.0 (perfect and strict hits identified only) was used (22). DIAMOND (23) v.0.8.36, or the Basic Local Alignment Search Tool (BLAST) (57) v.2.9.0 (where stated), was used to perform local alignment of Prodigal-predicted genes within contigs against CARD v.3.1.0 (22, 58). For metagenome assembled contigs, the *–low_quality* flag in RGI *main* was specified to allow prediction of partial open reading frames by Prodigal.

To predict ARGs from raw reads, KMA (41) v.1.3.8 within RGI *bwt* v.5.2.0 was used to align reads to CARD v.3.1.0. To predict resistance-conferring SNPs in ARGs detected with CARD's protein variant model, the consensus sequences generated from these read alignments were extracted and RGI *main* v.5.2.0 was used to predict SNPs, as described above.

**Quantification of antimicrobial resistance determinants and associated detection frequencies.** We consider ARGs as all genetic determinants of resistance, including resistance gene sequences and protein variants resulting from any mutation known to confer antimicrobial resistance. SNP identities were considered and analyzed in the protein variants in CARD. To quantify the occurrence of SNPs detected through CARD's protein variant model, the individual accession numbers and SNP identities for each gene from all RGI output files were extracted. For all other antimicrobial resistance determinants, unique ARGs were extracted from the "Best_Hit_ARO" (contigs) or "ARO_Term" (raw reads) column of each sample RGI output file, to create a new "unique AMR genes" file for each sample. ARG or SNP presence in a bootstrap sample was indicated by 1 and absence indicated by 0. Then, using Metaphlan2 v.2.9.14, the *merge_metaphlan_tables.py* was used to merge the "unique AMR genes" files together, where the first column outlined the ARGs predicted for all samples and the first row indicated the sample names. Merging the RGI output files allowed for the ARG detection frequency quantification. Briefly, detection frequencies for individual ARGs or SNPs were quantified by summing all the bootstraps within a specific subsample where the ARG or SNP was present and converting the total to a percentage. For example, in the *E. coli* ST38 genome, *bla*$_{CTX-M-15}$ was present in 60/100 bootstraps at a subsampling depth of 100,000 reads, thus the detection frequency of *bla*$_{CTX-M-15}$ is 60%. ARG detection frequencies across subsamples were visualized with GraphPad Prism v.9.3.0.

**ARG and SNP reference set based detection frequency analyses.** ARG and SNP reference sets are listed in Table S3. Where stated, detection frequencies of ARGs or SNPs present in the reference set were quantified. For analyses in the *E. coli* ST38 genome (100% relative abundance), reference #1 was used and consisted of all ARGs (*n* = 72) predicted from a single bootstrap of 5,000,000 *E. coli* ST38 contig-assembled reads (~250× coverage). For analyses in the combined metagenomic samples, reference #2 was used and consisted of ARGs unique to reference #1 (*n* = 20) and not predicted in contigs assembled from 82.5 million

(2 × 150 bp) microbial consortia reads. For SNP analyses, reference #3 was used and consisted of protein variants and associated SNPs predicted with CARD's protein variant model ($n = 6$), which were detected in a single bootstrap of 5,000,000 *E. coli* ST38 contig-assembled reads. To quantify additional ARGs in the combined metagenomic samples not found in the *E. coli* ST38 genome or the microbial consortia, reference #4 was used and consisted of all ARGs in reference #1 ($n = 72$) as well as microbial consortia specific ARGs ($n = 19$) predicted from 82.5 million reads assembled into contigs. The total ARGs in reference #4 is 90. Reference-based ARG and SNPs were plotted across subsamples, and minimum subsampling depths for individual ARGs and SNPs were summarized by three detection frequency cutoffs of greater than or equal to 50%, 90%, and 99% and visualized using R version 4.1.2 with the ggplot2 package.

**Performance analyses.** The performance of ARG detection was calculated using sensitivity and PPV or recall and precision. To calculate sensitivity and PPV across subsamples for the *E. coli* ST38 isolate, the ARGs predicted in each bootstrap of a subsample using a contig assembly approach with SPAdes or a read-based approach with KMA were compared to reference set #1 for ARGs (Table S3). If the ARG was present in the bootstrap and reference, the gene was considered a true positive. If the ARG was not present in neither the bootstrap nor the reference, this gene was considered a true negative. False positive ARGs were present in the bootstrap but absent in the reference and false negative ARGs were absent in the bootstrap but present in the reference. For each bootstrap sample, the true positives, true negatives, false positives, and false negatives were summed at each subsample, and sensitivity and PPV or recall and precision were calculated.

**Coverage estimation.** Sequencing coverage was estimated using the Lander-Waterman equation (59). We overestimated *E. coli* ST38's genome size by determining the value which fell within one standard deviation of the mean genome size of all complete *E. coli* genomes deposited in NCBI (5.6 Mbp) and rounding to 6 Mbp, which represents the larger size of *E. coli* genomes found in NCBI's microbial genome database. To estimate the number of reads required to detect *E. coli* ST38 at a range of relative abundances, the minimum read requirement (300,000 reads) was divided by the target relative abundance. For example, if the target relative abundance was 10%, 300,000/0.10 would equal 3,000,000 reads.

**Validation from external data sets.** To validate the performance of a 300,000 read depth across a set of *E. coli* isolates (24), 300,000 reads were subsampled, once, from each isolate, assessed for quality with FastQC. Isolates were discarded if they failed per base sequence quality. We then compared the ARGs detected at 300,000 reads to the ARGs detected from the original sequence depth. The true positives, true negatives, false positives, and false negatives were summed for each isolate, then sensitivity, PPV, and F1 score as a balanced measure of sensitivity and PPV were calculated.

To target 15× target genome coverage in metagenomic samples (26) and to demonstrate *vanA* detection frequency across a range of *Enterococcus* genome coverages (0.5×–15×), we subsampled each metagenomic sample and bootstrapped each subsample 10 times. Each subsample depth was calculated using the Lander-Waterman equation, as described above, while accounting for the *Enterococcus* relative abundance in the sample, as determined using Metaphlan2. Similarly to the *E. coli* genome estimation described above, we overestimated the *Enterococcus* genome size at 4 Mbp.

**ARG detection frequency assessment of published data sets.** Post-quality filtered sequence depths were extracted that were provided in each study's supplementary material for Studies 1 (9), 2 (6), and 3 (10). For Study 3, the sequences reported under the heading "After human contamination removal" under the subheading "read-pairs" were used. Sequencing read lengths were reported in Study 2 (2 × 150 bp) and 3 (2 × 100 bp), but for Study 1 we extracted the read lengths from the individual studies referenced within the paper. For each sample, coverage was estimated from the published data sets and for each subsample performed on the samples where *E. coli* ST38 represented 100%, 90%, 50%, 10%, and 1% relative abundance, assuming a genome length of 6 Mbp, then these values were log-transformed. Using GraphPad Prism version 9.1.2, sigmoidal curves were fit to detection frequency data for the $bla_{CTX-M-15}$ for each sample, where *E. coli* ST38 represented 100%, 90%, 50%, 10%, and 1% relative abundance. The equations were constrained at 0 and 100, and detection frequency was interpolated for relative abundances 100%, 90%, 50%, 10%, and 1% based on coverage estimation.

**Statistical analyses.** Comparisons between paired ARG detection frequencies using KMA or SPAdes/metaSPAdes assemblies for ARG prediction were made using the Wilcoxon matched-pairs signed-rank test at each subsample in GraphPad Prism v.9.3.0.

**Data availability.** The data set generated during the current study is available in the NCBI sequence read archive under BioProject PRJNA649958. Metadata for the generated samples can be found in Table S4. The *Escherichia coli* genome set analyzed in this study can be found at NCBI under BioProject PRJNA521038, and through Zenodo.org under the DOIs 10.5281/zenodo.3706855 and 10.5281/zenodo.3701595. Sequencing data from a public data set of 10 rectal swab samples analyzed in this study can be found in the NCBI sequence read archive under BioProject PRJNA655185.

## SUPPLEMENTAL MATERIAL

Supplemental material is available online only.

**DATA SET S1**, XLSX file, 0.04 MB.
**DATA SET S2**, XLSX file, 0.01 MB.
**DATA SET S3**, XLSX file, 0.02 MB.
**FIG S1**, PDF file, 1.1 MB.
**FIG S2**, PDF file, 1.3 MB.
**FIG S3**, PDF file, 0.3 MB.

**TABLE S1**, XLSX file, 0.02 MB.
**TABLE S2**, XLSX file, 0.01 MB.
**TABLE S3**, XLSX file, 0.01 MB.
**TABLE S4**, XLSX file, 0.01 MB.

## ACKNOWLEDGMENTS

This research was funded by Ontario Genomics SPARK program, PSI Foundation.

Collaborative Health Research Program Project Grant from the Canadian Institutes for Health Research and the Natural Sciences and Engineering Research Council of Canada (CP-151952), Canadian Foundation for Innovation, and McLaughlin Accelerator Grant to B.C. Equipment was provided to B.C. by the Canadian Foundation for Innovation John Evans Leadership Fund. In addition, funding was provided by the Canadian Institutes of Health Research (PJT-156214 to A.M.) and a David Braley Chair in Computational Biology to A.M. Some computer resources were supplied by the McMaster Service Lab and Repository computing cluster, funded in part by grants to A.M. from the Canada Foundation for Innovation (34531) and hardware donations from Cisco Systems Canada, Inc.

We greatly appreciate Emma Allen-Vercoe from the University of Guelph for sending us aliquots of the microbial consortia.

A.M.R., P.H.H.S., and B.C. conceived and designed the study. B.C. supervised the overall study. A.M.R. with the guidance and supervision of P.H.H.S. performed sample preparation and bioinformatic analyses. A.R.R. provided feedback on and performed some bioinformatic analyses. R.G.M. and C.S. provided qPCR and Sanger sequencing support. R.G.M., D.R.M., and A.G.M. provided feedback during analyses. N.R.Y. provided help with the bioinformatic analyses. A.M.R. generated the figures and wrote the manuscript. All authors provided feedback during manuscript preparation and have read and approved the final manuscript.

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
