## [Reviewer comments · mSystems]

Performance characteristics of next-generation sequencing for the detection of antimicrobial resistance determinants in *Escherichia coli* genomes and metagenomes

Ashley Rooney, Amogelang Raphenya, Roberto Melano, Christine Seah, Noelle Yee, Derek MacFadden, Andrew McArthur, Pierre Schneeberger, and Bryan Coburn

Corresponding Author(s): Bryan Coburn, University Health Network, University Health Network and Toronto General Hospital Research Institute

Review Timeline:

Submission Date:	January 6, 2022
Editorial Decision:	March 4, 2022
Revision Received:	April 14, 2022
Accepted:	May 4, 2022

Editor: Charles Langelier

Reviewer(s): Disclosure of reviewer identity is with reference to reviewer comments included in decision letter(s). The following individuals involved in review of your submission have agreed to reveal their identity: Xian-Zhi Li (Reviewer #2)

Transaction Report:

DOI: <https://doi.org/10.1128/msystems.00022-22>

March 4, 2022

Dr. Bryan Coburn
University Health Network, University Health Network and Toronto General Hospital Research Institute
Department of Medicine, University of Toronto
Toronto
Canada

Re: mSystems00022-22 (Performance characteristics of next-generation sequencing for the detection of antimicrobial resistance determinants in *Escherichia coli* genomes and metagenomes)

Dear Dr. Bryan Coburn:

Thank you for submitting your manuscript to mSystems. We have completed our review and I am pleased to inform you that, in principle, we expect to accept it for publication in mSystems. However, acceptance will not be final until you have adequately addressed the reviewer comments.

Preparing Revision Guidelines

Sincerely,

Charles Langelier

Editor, mSystems

Journals Department
American Society for Microbiology

Reviewer comments:

Reviewer #2 (Comments for the Author):

This ms reports the performance of whole genome sequence based approaches to detect antimicrobial resistance genes (ARGs) in single isolates and in metagenomics samples. The authors constructed a simulated metagenomic sample by spiking a characterized multidrug resistant *E. coli* ST38 strain at varying abundances, and then assessed the ability of assembly-based (Spades) and -free (read-based KMA) methods to detect ARGs. The findings reveal that a minimum of 15X genome coverage was needed to detect ARGs in *E. coli* single genomes. This coverage was also sufficient to detect *E. coli* ARGs in metagenomic samples. However, in a metagenomics sample, the number of *E. coli* sequences must increase proportionally to the decrease in relative abundance. The data also demonstrated that the assembly method performed better than the read-based approach (KMA). Overall, the experiments and analyses have been well performed, resulting in important observations. Limitations of the approach are also presented. Several specific comments are give below.

Generally, reads based approaches such as KMA enable identification of ARGs from low-abundance organisms present in complex communities, which may be missed by assembly-based methods owing to incomplete or poor assemblies. However, the authors have showed that compared to the assembly approach, KMA did not improve the limit of detection for resistance conferring SNPs and also predicted non-reference ARGs with increasing read depths (Fig 3 h). Could these observations be attributed to the database selection and/or the read alignment tool used?

According to Fig 2 f, as the relative abundance decreases more sequencing effort is needed to achieve a 100% detection frequency. Can the authors comment on the sequence quality and read length of the samples used in this analysis?

The reviewer would recommend that the ms include in the supplementary materials the assembly quality metrics of the subsample sequencing reads used to simulate sequencing at lower depths in the *E. coli* ST38 genome.

L143L/192. "46 efflux genes" (Fig 1 b - orange for efflux genes)/"90.5% (19/21)". As, totally expected, a large number efflux genes are noted. However, the description of "efflux" may need clarity. The reviewer hopes to see the exact list of these genes for clarity because various resistance gene database may indifferentially include regulatory genes (whose products are linked with efflux transporters genes) and other genes that do not code efflux pump components of the transporter systems such as outer membrane components (e.g., TolC). For example, in the supplementary excel sheet, the reviewer notes the presence of many regulatory genes such as those coding for SoxS, SoxR, MarA, EvgA, EvgS, BaeS, BaeR, CRP, EmrR, etc, which would not be qualified as efflux genes per se. Their presence or not often does not link to resistance phenotypes at all, which is different from the identification of genes for beta-lactamases or aminoglycoside modifying enzymes. Thus, the ms would benefit from providing additional descriptions on those "efflux" genes.

Minors: L89/L117/L118. Spell out "AGRs", "PPV" and "SNP" for their first appearance in the main body of text.

L166/167. Write "pulfomycin", "fosfomycin" (lower case "P" and "F").

L171-172. "...SNPs, except for the *gyrA* where the resistance-conferring SNPs were detected 1-2% less frequently than the variant itself (Supplementary Table3)". There is no legend to Supplementary Table 3 and "1-2%" less detection of *gyrA* is not indicated in the table.

Reviewer #3 (Comments for the Author):

The study provides an analysis on the sequencing depths required to detect antibiotic resistance genes (ARGs) in metagenomes using 150 bp Illumina reads. The conclusion that a minimum of 15X coverage is needed for the detection of AMR genes using an assembly-based approach is very useful, not only to standardize the analysis of metagenomes but also for the detection ARGs in isolated bacteria.

The manuscript indicates in several sentences "reads" but it lacks details on the size and quality of these reads. Since there are numerous sequencing platforms that produce reads of different sizes (2x75 bp, 2x100 bp, 2x150 bp, 2x250 bp, 2x300 bp) and qualities (Illumina vs. PacBio, Oxford Nanopore), it is critical to specify the type of "reads"; see lines 53, 56, 72, 108, etc. If the term "reads" in the manuscript refers to Illumina reads that are 150 bp, it needs to be specified at the beginning of the manuscript.

The readability of the manuscript would improve by stating the objectives of the study at the end of the introduction. The introduction does not mention the study on the detection of vanA from *Enterococcus* in the 10 rectal surveillance swabs and the detection of ARG in published data sets (these are large sections of the results).

The introduction should briefly explain the two approaches used to detect ARG in metagenomes: read-based and assembly-based.

For rigor and reproducibility, please provide the version and parameters used for all software. If default parameters were used, it needs to be specified for each software used.

Please provide detail on how the reads were quality controlled (e.g. what percentage of reads used in the analysis had Q>20 and what percentage had Q>30?). Lines 441-442 indicates "Paired-end fastq files (read 1 and read 2) were assessed for quality using FastQC (48)" but it did not specify the minimum read quality and length needed to be included in the study.

Line 322 indicates that "this study precisely quantifies sequencing depths required to detect ARGs in a target organism across varying relative abundances in mixed metagenomes using standard methods". The statement is not accurate because the method used (screening metagenomes) cannot reveal the organism that carries the ARG; i.e. the method does not allow to link an ARG to a target organism because the DNA of all bacteria are mixed in a metagenome. It is well known that the same ARGs can be present in different species of bacteria by horizontal gene transfer.

The manuscript should specify whether other sequence types of *E. coli* are present in microbial consortia and discuss how the presence of additional strains of susceptible *E. coli* would change the detection of ARGs; i.e. mixing alleles (e.g. *gyrA*) with and without point mutations at different strain proportions (e.g. 99:1, 90:10, 50:50, 10:90, 1:99 ratios of susceptible and resistant strains of the same species). If the effects of having mixes of susceptible and resistant strains of the same species on the detection of ARGs is unknown, it should be stated as a limitation.

Lines 388-390 indicate that ". . . sequencing-based approaches may be augmented with bait capture approaches that enrich for ARG targets, although these currently cannot capture resistance-conferring SNPs, but may increase efficiency and sensitivity". If a SNP do not confer resistance what is its clinical significance? If a SNP does not confer resistance, should it be included as an ARG?

Line 414-415 indicates ". . . resistance conferring SNPs in *parC* (S80I) and *gyrA* (S83L, D87N)" but it does not indicate to the antibiotic that these point mutations confer resistance to. Please specify the antibiotic that these point mutations confer resistance to (fluorquinolones).

The term "microbial consortia" is introduced in the discussion (line 366), but it is not mentioned in the results. The results should start explaining that *E. coli* ST38 isolate was spiked to the microbial consortia because "each isolate included in the mixture underwent antibiotic susceptibility testing and was included if phenotypically susceptible to a range of antibiotics" from lines 374-375.

In Fig 2C, *cyaA*, *gyrA* and *parC* are resistance conferring SNPs, but they are marked as ARGs. What was the criteria used to classify a gene as SNP or as ARGs?

The study provides an analysis on the sequencing depths required to detect antibiotic resistance genes (ARGs) in metagenomes using 150 bp Illumina reads. The conclusion that a minimum of 15X coverage is needed for the detection of AMR genes using an assembly-based approach is very useful, not only to standardize the analysis of metagenomes but also for the detection ARGs in isolated bacteria.

The manuscript indicates in several sentences “reads” but it lacks details on the size and quality of these reads. Since there are numerous sequencing platforms that produce reads of different sizes (2x75 bp, 2x100 bp, 2x150 bp, 2x250 bp, 2x300 bp) and qualities (Illumina vs. PacBio, Oxford Nanopore), it is critical to specify the type of “reads”; see lines 53, 56, 72, 108, etc. If the term “reads” in the manuscript refers to Illumina reads that are 150 bp, it needs to be specified at the beginning of the manuscript.

The readability of the manuscript would improve by stating the objectives of the study at the end of the introduction. The introduction does not mention the study on the detection of *vanA* from *Enterococcus* in the 10 rectal surveillance swabs and the detection of ARG in published data sets (these are large sections of the results).

The introduction should briefly explain the two approaches used to detect ARG in metagenomes: read-based and assembly-based.

For rigor and reproducibility, please provide the version and parameters used for all software. If default parameters were used, it needs to be specified for each software used.

Please provide detail on how the reads were quality controlled (e.g. what percentage of reads used in the analysis had $Q>20$ and what percentage had $Q>30$?). Lines 441-442 indicates “Paired-end fastq files (read 1 and read 2) were assessed for quality using FastQC (48)” but it did not specify the minimum read quality and length needed to be included in the study.

Line 322 indicates that “this study precisely quantifies sequencing depths required to detect ARGs in a target organism across varying relative abundances in mixed metagenomes using standard methods”. The statement is not accurate because the method used (screening metagenomes) cannot reveal the organism that carries the ARG; i.e. the method does not allow to link an ARG to a target organism because the DNA of all bacteria are mixed in a metagenome. It is well known that the same ARGs can be present in different species of bacteria by horizontal gene transfer.

The manuscript should specify whether other sequence types of *E coli* are present in microbial consortia and discuss how the presence of additional strains of susceptible *E coli* would change the detection of ARGs; i.e. mixing alleles (e.g. *gyrA*) with and without point mutations at different strain proportions (e.g. 99:1, 90:10, 50:50, 10:90, 1:99 ratios of susceptible and resistant strains of the same species). If the effects of having mixes of susceptible and resistant strains of the same species on the detection of ARGs is unknown, it should be stated as a limitation.

Lines 388-390 indicate that “. . . sequencing-based approaches may be augmented with bait capture approaches that enrich for ARG targets, although these currently cannot capture resistance-conferring SNPs, but may increase efficiency and sensitivity”. If a SNP do not confer resistance what is its clinical significance? If a SNP does not confer resistance, should it be included as an ARG?

Line 414-415 indicates “. . . resistance conferring SNPs in *parC* (S80I) and *gyrA* (S83L, D87N)” but it does not indicate to the antibiotic that these point mutations confer resistance to. Please specify the antibiotic that these point mutations confer resistance to (fluorquinolones).

The term “microbial consortia” is introduced in the discussion (line 366), but it is not mentioned in the results. The results should start explaining that *E. coli* ST38 isolate was spiked to the microbial consortia because “each isolate included in the mixture underwent antibiotic susceptibility testing and was included if phenotypically susceptible to a range of antibiotics” from lines 374-375.

In Fig 2C, *cyaA*, *gyrA* and *parC* are resistance conferring SNPs, but they are marked as ARGs. What was the criteria used to classify a gene as SNP or as ARGs?

Editors Comments

Thank you for submitting your manuscript to mSystems. We have completed our review and I am pleased to inform you that, in principle, we expect to accept it for publication in mSystems. However, acceptance will not be final until you have adequately addressed the reviewer comments.

Thank you for the privilege of reviewing your work

Thank-you for considering our manuscript. Please see below the responses to the reviewers and changes we have made to the manuscript, which we hope directly addresses the reviewers' comments to your and their satisfaction.

We have uploaded a revised manuscript with the changes highlighted and a final clean version.

Reviewer #2

This ms reports the performance of whole genome sequence based approaches to detect antimicrobial resistance genes (ARGs) in single isolates and in metagenomics samples. The authors constructed a simulated metagenomic sample by spiking a characterized multidrug resistant E. coli ST38 strain at varying abundances, and then assessed the ability of assembly-based (Spades) and -free (read-based KMA) methods to detect ARGs. The findings reveal that a minimum of 15X genome coverage was needed to detect ARGs in E. coli single genomes. This coverage was also sufficient to detect E. coli ARGs in metagenomic samples. However, in a metagenomics sample, the number of E. coli sequences must increase proportionally to the decrease in relative abundance. The data also demonstrated that the assembly method performed better than the read-based approach (KMA). Overall, the experiments and analyses have been well performed, resulting in important observations. Limitations of the approach are also presented. Several specific comments are given below.

We thank-you for taking the time to review our manuscript and appreciate your feedback. We have attempted to address your comments below.

1. Generally, reads based approaches such as KMA enable identification of ARGs from low-abundance organisms present in complex communities, which may be missed by assembly-based methods owing to incomplete or poor assemblies. However, the authors have showed that compared to the assembly approach, KMA did not improve the limit of detection for resistance conferring SNPs and also predicted non-reference ARGs with increasing read depths (Fig 3 h). Could these observations be attributed to the database selection and/or the read alignment tool used?

Response to Reviewer #2: Comment #1

Thank-you for your comment. For the analyses generated in Figure 3, it is more likely that KMA itself rather than the selection of an ARG database, which often contain similar sets of genes and alleles, affected the performance of ARG detection.

We have added an additional information in the discussion describing the issue of aligning sequences to redundant gene databases and the potential impact on ARG detection performance.

Changes in the manuscript: “..., where some genes (e.g. *gyrA* and *parC* SNPs and *bla_{TEM-1}*) that had a high detection frequency with assembly, had a low detection frequency with KMA. Although KMA was designed to overcome the problem of mapping sequences to highly redundant gene databases (40), our results suggest that KMA suffered still from the antimicrobial resistance allele network problem where one sequence can map to multiple alleles which can increase false positives (41). We found that the increased number of false positives detected with KMA could not be filtered without affecting the overall sensitivity of the assay. The differentiation of ARG alleles is important as different alleles, for example *mcr-1* versus *mcr-9* are associated with different resistance phenotypes (42).”(Line 382 – 392)

According to Fig 2 f, as the relative abundance decreases more sequencing effort is needed to achieve a 100% detection frequency. Can the authors comment on the sequence quality and read length of the samples used in this analysis?

Response to Reviewer #2: Comment #2

Thank-you for your comment. We have summarized sequence quality information for the samples sequenced in this study across subsamples in Supplementary Tables 6 and 7. Supplementary Table 6 summarizes the proportion of sequences with a Phred score of >Q20 as well as >Q30. As expected with Illumina sequencing data, close to 100% of sequences had a Phred score >Q20 and 98-99% of sequences had a Phred score >Q30.

We performed 2 X 150 base pair sequencing where 1-2% of the sequences in each sample across subsamples had less than 35 bp long. This information is summarized in Supplementary Table 7.

- 2. The reviewer would recommend that the ms include in the supplementary materials the assembly quality metrics of the subsample sequencing reads used to simulate sequencing at lower depths in the *E. coli* ST38 genome.**

Response to Reviewer #2: Comment #3

Thank-you for this suggestion. We have included Supplementary Figure 3 a – c which displays the total number of contigs assembled (Suppl Fig. 3a), the total length of the assembly in base pairs (bp) (Suppl Fig. 3b), and the N50 (Suppl Fig. 3c) for all sequenced samples across subsamples.

We updated the methods section to include the use of Quast v.5.2.0 to assess these contiguity-based metrics.

- 3. L143L/192. "46 efflux genes" (Fig 1 b - orange for efflux genes)/"90.5% (19/21)".** As, totally expected, a large number efflux genes are noted. However, the description of "efflux" may need clarity. The reviewer hopes to see the exact list of these genes for clarity because various resistance gene database may indifferentially include regulatory genes (whose products are linked with efflux transporters genes) and other genes that do not code efflux pump components of the transporter systems such as outer membrane components (e.g., TolC). For example, in the supplementary excel sheet, the reviewer notes the presence of many regulatory genes such as those coding for SoxS, SoxR, MarA, EvgA, EvgS, BaeS, BaeR, CRP, EmrR, etc, which would not be qualified as efflux genes per se. Their presence or not often does not link to resistance phenotypes at all, which is different from the identification of genes for beta-lactamases or aminoglycoside modifying enzymes. Thus, the ms would benefit from providing additional descriptions on those "efflux" genes.

Response to Reviewer #2: Comment #4

Thank-you for your comment. The ARGs the reviewer refers to in Figure 1b that are detected with $\geq 90\%$ detection frequency in relevant subsamples, can be found in Supplementary Table 1. We have added an additional column to this table which outlines the resistance mechanisms associated with each gene and an additional column describing whether the genes would be regulatory or not.

ARGs included in the CARD, including mutated regulatory genes, have been rigorously curated and are included in the database if there is demonstrated evidence in peer-reviewed publications that the ARG elevates MICs.

- 4. Minors: L89/L117/L118. Spell out "AGRs", "PPV" and "SNP" for their first appearance in the main body of text.**

Response to Reviewer #2: Comment #5

Thank-you, we have spelled out the first appearance of acronyms in the main body of the text and ensured others were spelled out as well.

- 5. L166/167. Write "pulvomycin", "fosfomycin" (lower case "P" and "F").**

Response to Reviewer #2: Comment #6

Thank-you we have made appropriate changes.

6. L171-172. "...SNPs, except for the *gyrA* where the resistance-conferring SNPs were detected 1-2% less frequently than the variant itself (Supplementary Table3)". There is no legend to Supplementary Table 3 and "1-2%" less detection of *gyrA* is not indicated in the table.

Response to Reviewer #2: Comment #7

Thank-you, in Supplementary Table 3., we have added a note under the title. **"Note: "Any" refers to the detection of the gene variant itself, without considering the SNPs that confer resistance. All SNPs detected for each gene variant are listed. The SNPs previously characterized in *gyrA* and *parC* that confer resistance to fluoroquinolones are D87N, S83L and S80I, respectively."**

We have also updated the manuscript to provide more clarification.

Changes in the manuscript: "...except for *gyrA* with the resistance-conferring SNPs D87N, S83L where these SNPs were detected 1-2% less frequently than the variant itself at subsamples 50k to 150k. Instead S83L or D87N were detected individually (Supplementary Table 3)." (Lines 184 - 186).

Reviewer #3:

The study provides an analysis on the sequencing depths required to detect antibiotic resistance genes (ARGs) in metagenomes using 150 bp Illumina reads. The conclusion that a minimum of 15X coverage is needed for the detection of AMR genes using an assembly-based approach is very useful, not only to standardize the analysis of metagenomes but also for the detection ARGs in isolated bacteria.

Thank-you for your comment. We agree that this study is useful, we hope that it will improve benchmarking studies and the interpretation of future studies assessing ARG content in a metagenome.

1. **The manuscript indicates in several sentences "reads" but it lacks details on the size and quality of these reads. Since there are numerous sequencing platforms that produce reads of different sizes (2x75 bp, 2x100 bp, 2x150 bp, 2x250 bp, 2x300 bp) and qualities (Illumina vs. PacBio, Oxford Nanopore), it is critical to specify the type of "reads"; see lines 53, 56, 72, 108, etc. If the term "reads" in the manuscript refers to Illumina reads that are 150 bp, it needs to be specified at the beginning of the manuscript.**

Response to Reviewer #3: Comment #1

Thank-you, we refer to reads in the abstract and results section, we included 2 X 150 bp in the abstract and results section where appropriate.

- 2. The readability of the manuscript would improve by stating the objectives of the study at the end of the introduction. The introduction does not mention the study on the detection of *vanA* from *Enterococcus* in the 10 rectal surveillance swabs and the detection of ARG in published data sets (these are large sections of the results).**

Response to Reviewer #3: Comment #2

Thank-you, based on your comment we have improved the introduction by including more information about the objectives of the study.

Changes in the manuscript: “The sequencing depths for the detection of all ARGs were validated in 948 *E. coli* genomes. For metagenomic samples, sequencing depths were validated for the detection of *vanA* in a publicly available dataset of 10 rectal swab samples with a range of *vanA*-carrying *Enterococcus* relative abundances.” (Lines 126 - 129)

- 3. The introduction should briefly explain the two approaches used to detect ARG in metagenomes: read-based and assembly-based.**

Response to Reviewer #3: Comment #3

Thank-you, we have addressed this comment by explaining assembly-based and read-based methods of ARG detection in the introduction.

Changes in the manuscript: “Using an assembly-based approach, ARGs can be predicted from genomes and metagenomes by performing *de novo* assembly (assembly without a reference) of the raw sequencing reads into contiguous sequences (contigs) and aligning the contigs to an ARG reference database. In contrast, a read-based approach directly aligns the raw genomic or metagenomic sequences to an ARG reference database for ARG prediction (17).” (Lines 101 – 106)

- 4. For rigor and reproducibility, please provide the version and parameters used for all software. If default parameters were used, it needs to be specified for each software used.**

Response to Reviewer #3: Comment #4

Thank-you for your comment, we have updated the methods section, under sub-heading “Bioinformatic analyses” to ensure all software had correct version numbers. We stated at the beginning of this section that “All bioinformatic analyses were performed with default settings except where stated.” and described any non-default parameters used.

- 5. Please provide detail on how the reads were quality controlled (e.g. what percentage of reads used in the analysis had Q>20 and what percentage had Q>30?). Lines 441-**

442 indicates "Paired-end fastq files (read 1 and read 2) were assessed for quality using FastQC (48)" but it did not specify the minimum read quality and length needed to be included in the study.

Response to Reviewer #3: Comment #5

Thank-you. The quality of the sequences was high, so no trimming or removal of short reads was applied. We have summarized read quality information in Supplementary table 6 which describes the average number of reads (%) that have a quality score of >Q20 as well as >Q30. Supplementary table 7 summarizes the number of very short reads (%) that are less than 20bp or 35 bp in length.

- 6. Line 322 indicates that "this study precisely quantifies sequencing depths required to detect ARGs in a target organism across varying relative abundances in mixed metagenomes using standard methods". The statement is not accurate because the method used (screening metagenomes) cannot reveal the organism that carries the ARG; i.e. the method does not allow to link an ARG to a target organism because the DNA of all bacteria are mixed in a metagenome. It is well known that the same ARGs can be present in different species of bacteria by horizontal gene transfer.**

Response to Reviewer #3: Comment #6

Thank-you, we appreciate this comment. We have updated the sentence you referred to and discussed this point in a few lines at the end of the discussion.

Changes to the manuscript: "...required to detect reference-associated ARGs across a range of relative abundances in mixed metagenomes using standard methods." (Line 343 - 344)

"We did not directly attempt to connect the ARGs detected in the metagenomic samples to the *E. coli* ST38 using bioinformatic techniques such as contig binning as it has been shown to produce false negatives for plasmid-borne ARGs (48). DNA manipulation methods such as Hi-C may be useful for linking ARGs to the bacterial hosts (49), but this method would require further validation." (Line 426 - 430)

- 7. The manuscript should specify whether other sequence types of *E. coli* are present in microbial consortia and discuss how the presence of additional strains of susceptible *E. coli* would change the detection of ARGs; i.e. mixing alleles (e.g. *gyrA*) with and without point mutations at different strain proportions (e.g. 99:1, 90:10, 50:50, 10:90, 1:99 ratios of susceptible and resistant strains of the same species). If the effects of having mixes of susceptible and resistant strains of the same species on the detection of ARGs is unknown, it should be stated as a limitation.**

Response to Reviewer #3: Comment #7

Thank-you, this is an important point, which is currently an active area of research for benchmarking a range of bioinformatic tools for metagenome analyses. This is worth discussing as a limitation.

We have included a sentence in the results section (Line 213 - 214) indicating the presence of a susceptible *E. coli* in the microbial consortium.

Changes in the manuscript: “Even in the presence of a susceptible *E. coli* likely carrying wild-type alleles in the microbial consortia, we found that we could detect the *E. coli* ST38 *gyrA* and *parC* protein variants including their associated SNPs provided the minimum sequencing depth was achieved. However, there were challenges detecting the *gyrA* variant in the combined metagenomic sample where the spiked-in *E. coli* ST38 accounted for 1% of the total sample and the susceptible *E. coli* accounted for 2% of the total sample. A total of 30 million reads allowed for the detection of the mutant *parC* with its associated SNPs, but we did not detect the mutant *gyrA* variant even at this high sequencing depth. Our sample set was not created to directly assess the performance of detecting protein variants (e.g *gyrA* or *parC*) in the presence of more abundant wild-type genes. “ (Line 393 - 403).

- 8. Lines 388-390 indicate that “. . . sequencing-based approaches may be augmented with bait capture approaches that enrich for ARG targets, although these currently cannot capture resistance-conferring SNPs, but may increase efficiency and sensitivity”. If a SNP do not confer resistance what is its clinical significance? If a SNP does not confer resistance, should it be included as an ARG?**

Response to Reviewer #3: Comment #8

Thank-you, you are correct SNPs that do not confer resistance should not be included as an ARG. CARD only includes SNPs in protein variants that have been proven to elevate MICs. Thus any SNPs we detect using the CARD database are resistance-conferring.

- 9. Line 414-415 indicates “. . . resistance conferring SNPs in *parC* (S80I) and *gyrA* (S83L, D87N)” but it does not indicate to the antibiotic that these point mutations confer resistance to. Please specify the antibiotic that these point mutations confer resistance to (fluorquinolones).**

Response to Reviewer #3: Comment #9

Thank-you, we have made the appropriate changes.

- 10. The term “microbial consortia” is introduced in the discussion (line 366), but it is not mentioned in the results. The results should start explaining that *E. coli* ST38 isolate was spiked to the microbial consortia because “each isolate included in the mixture underwent antibiotic susceptibility testing and was included if phenotypically susceptible to a range of antibiotics” from lines 374-375.**

Response to Reviewer #3: Comment #10

Thank-you, we have included this information at the beginning of the sub-heading “Detection of *E. coli* ST38 ARGs and SNPs at a range of relative abundances in a metagenomic sample” in the results section.

Changes in the manuscript: “...where each isolate was subject to antibiotic susceptibility testing and included in the consortium if phenotypically susceptible to a range of antibiotics,” (Lines 210 - 211).

11. In Fig 2C, *cyaA*, *gyrA* and *parC* are resistance conferring SNPs, but they are marked as ARGs. What was the criteria used to classify a gene as SNP or as ARGs?

Response to Reviewer #3: Comment #11

Thank-you for your comment. We have added an additional sentence in the methods section for clarification.

Changes in the manuscript: “...We consider ARGs as all genetic determinants of resistance, including resistance gene sequences and protein variants resulting from any mutation known to confer antimicrobial resistance. SNP identities were considered and analyzed in the protein variants in CARD.” (Line 504 - 506)

May 4, 2022

Dr. Bryan Coburn
University Health Network, University Health Network and Toronto General Hospital Research Institute
Department of Medicine, University of Toronto
Toronto
Canada

Re: mSystems00022-22R1 (Performance characteristics of next-generation sequencing for the detection of antimicrobial resistance determinants in *Escherichia coli* genomes and metagenomes)

Dear Dr. Coburn:

Thank you for your manuscript revisions. I'm pleased to let you know that we've now officially accepted it and I am forwarding it to the ASM Journals Department for publication. For your reference, ASM Journals' address is given below. Before it can be scheduled for publication, your manuscript will be checked by the mSystems production staff to make sure that all elements meet the technical requirements for publication. They will contact you if anything needs to be revised before copyediting and production can begin. Otherwise, you will be notified when your proofs are ready to be viewed.

Publication Fees:

We recognize that the video files can become quite large, and so to avoid quality loss ASM suggests sending the video file via <https://www.wetransfer.com/>. When you have a final version of the video and the still ready to share, please send it to mSystems staff at mssystems@asmusa.org.

For mSystems research articles, if you would like to submit an image for consideration as the Featured Image for an issue, please contact mSystems staff at mssystems@asmusa.org.

Sincerely,

Charles Langelier
Editor, mSystems

Journals Department
Supplementary Figure 3: Accept
Table S3: Accept
Data Set S2: Accept
Data Set S3: Accept
Supplementary Figure 2: Accept
Table S4: Accept
Data Set S1: Accept
Supplementary Figure 1: Accept
Table S1: Accept
Table S2: Accept